# Direct inference and control of genetic population structure from RNA sequencing data

Muhamad Fachrul [1,2,3✉], Abhilasha Karkey[4,5], Mila Shakya[4,5], Louise M. Judd[6], Taylor Harshegyi [6], Kar Seng Sim[7], Susan Tonks[8], Sabina Dongol[4,5], Rajendra Shrestha[5], Agus Salim[9,10,11], STRATAA study group*, Stephen Baker[12], Andrew J. Pollard [8], Chiea Chuen Khor [7], Christiane Dolecek[13,14], Buddha Basnyat[4,13], Sarah J. Dunstan [15], Kathryn E. Holt [6,16] & Michael Inouye [1,2,17,18,19,20,21✉]

RNAseq data can be used to infer genetic variants, yet its use for estimating genetic population structure remains underexplored. Here, we construct a freely available computational tool (RGStraP) to estimate RNAseq-based genetic principal components (RG-PCs) and assess whether RG-PCs can be used to control for population structure in gene expression analyses. Using whole blood samples from understudied Nepalese populations and the Geuvadis study, we show that RG-PCs had comparable results to paired array-based genotypes, with high genotype concordance and high correlations of genetic principal components, capturing subpopulations within the dataset. In differential gene expression analysis, we found that inclusion of RG-PCs as covariates reduced test statistic inflation. Our paper demonstrates that genetic population structure can be directly inferred and controlled for using RNAseq data, thus facilitating improved retrospective and future analyses of transcriptomic data.

[1] Cambridge Baker Systems Genomics Initiative, Baker Heart and Diabetes Institute, Melbourne, VIC, Australia. [2] Department of Clinical Pathology, University of Melbourne, Parkville, VIC, Australia. [3] School of BioSciences, The University of Melbourne, Parkville, VIC, Australia. [4] Oxford University Clinical Research Unit, Patan Academy of Health Sciences, Kathmandu, Nepal. [5] Patan Academy of Health Sciences, Patan Hospital, Lalitpur, Nepal. [6] Department of Infectious Diseases, Central Clinical School, Monash University, Melbourne, VIC, Australia. [7] Genome Institute of Singapore, Singapore, Singapore. [8] Oxford Vaccine Group, Department of Paediatrics, University of Oxford, and the NIHR Oxford Biomedical Research Centre, Oxford, UK. [9] Centre for Epidemiology and Biostatistics, Melbourne School of Population and Global Health, The University of Melbourne, Melbourne, VIC, Australia. [10] School of Mathematics and Statistics, The University of Melbourne, Melbourne, VIC, Australia. [11] Department of Population Health, Baker Heart and Diabetes Institute, Melbourne, VIC, Australia. [12] Department of Medicine, University of Cambridge, Cambridge, UK. [13] Nuffield Department of Medicine, Centre for Tropical Medicine and Global Health, University of Oxford, Oxford, UK. [14] Mahidol Oxford Tropical Medicine Research Unit, Mahidol University, Bangkok, Thailand. [15] The Peter Doherty Institute for Infection and Immunity, The University of Melbourne, Melbourne, VIC, Australia. [16] Department of Infection Biology, London School of Hygiene & Tropical Medicine, London, UK. [17] Cambridge Baker Systems Genomics Initiative, Department of Public Health and Primary Care, University of Cambridge, Cambridge, UK. [18] Health Data Research UK Cambridge, Wellcome Genome Campus and University of Cambridge, Cambridge, UK. [19] British Heart Foundation Cardiovascular Epidemiology Unit, Department of Public Health and Primary Care, University of Cambridge, Cambridge, UK. [20] British Heart Foundation Centre of Research Excellence, University of Cambridge, Cambridge, UK. [21] Victor Phillip Dahdaleh Heart and Lung Research Institute, University of Cambridge, Cambridge, UK. *A list of authors and their affiliations appears at the end of the paper. ✉email: fachrul.mhmd@gmail.com; mi336@medschl.cam.ac.uk

RNA sequencing (RNAseq) has revolutionized our understanding of the transcriptome, offering both accurate quantification method for gene expression as well as identification of specific alternative splicing sites and cell-type-specific transcripts[1,2]. Its application extends to the clinical setting, allowing us to further elucidate complex diseases and identify prospective biomarkers in both communicable and non-communicable diseases[3].

Yet, studies using RNAseq rarely consider the germline genetic variation also contained within RNAseq read sets. Studies which do not leverage this information may be vulnerable to bias and confounding, such as population stratification, which may affect transcription between groups[4–7]. To overcome this issue, researchers have typically relied on genome-wide array or whole genome sequence (WGS) data matched for the same individuals with RNAseq. This allows researchers to deploy approaches to control for population stratification, such as the calculation of genetic principal components (PCs) and their use as covariates in subsequent statistical association models[8–10]. The genetic PCs are taken to represent the latent genetic structure within and between populations, which introduce confounding due to differences in social environment[11] or (in the case of differential gene expression) due to heterogeneity of quantitative trait loci between groups. However, the need for genome-wide array or WGS to match with RNAseq data is potentially unnecessary and indeed may not be possible in settings where resources are limited, such as Low and Low Middle Income Countries (LMICs) with highly diverse and understudied populations.

It has been demonstrated that genotype calls can be made from RNAseq data using tools such as GATK[12–14]. The approach of utilizing RNAseq data to capture genetic structure has been applied for livestock and agricultural purposes[15–18], for example to investigate the population structure, history and adaptation of domesticated barley (*Hordeum vulgare*)[17]. While proof-of-concept and subsequent utility of RNAseq-based genotypes have been demonstrated such as for tissue-specific variants[19], its application to infer human population structure shows promise yet remains relatively underexplored[20].

The aims of this study are to (i) demonstrate that RNAseq-based genotypes can capture the genetic population structure of a diverse yet understudied human population, and (ii) show that use of RNAseq-based genetic principal components (RG-PCs) can effectively control for population structure in association analysis. Here, we recruited and generated whole blood RNAseq data of 376 individuals from Nepal, a landlocked country situated in the Himalayas with over 125 ethnic groups[21,22]. We developed an RNAseq analysis pipeline (RGStraP) to calculate genetic principal components directly from RNAseq data, then validated RGStraP's performance with genome-wide array genotype data from the same Nepalese individuals. We also tested the pipeline on samples from the Geuvadis consortium, which contains 465 samples with paired genotype-RNAseq data from five of the 1000 Genomes populations[23]. Finally, we show the validity of adjusting for RG-PCs in an association analysis to identify sex-specific gene expression. Overall, our study establishes that human population structure, particularly from an understudied but diverse population, can be effectively captured and controlled directly using RNAseq data.

## Results

In this study, we constructed the RGStraP pipeline to calculate RG-PCs from genetic variants called from RNAseq data. RGStraP relies on GATK for its variant calling suite, as well as PLINK and flashPCA to filter the SNPs and calculate genetic principal components from them, respectively (Methods). We make RGStraP available to the community via github (https://github.com/fachrulm/RGStraP)[24].

**RNAseq-based variant calling captures comparable population structure to paired array genotypes**. We collected whole blood samples from 376 individuals recruited in Latlipur, Nepal as part of the STRATAA study, then performed whole blood RNAseq using Illumina Novaseq (Methods). The cohort included individuals with and without confirmed S. Typhi infection; for the purpose of this study, the disease groups were used as a regression covariate to adjust for gene expression during downstream analyses. Self-reported ethnicity showed individuals belonging to 6 broad ethnic / caste groups.

We ran the RGStraP pipeline (Fig. 1) on 376 whole blood RNAseq samples, of which 362 passed QC after genetic variant calling. A total of 4,921,472 genetic variants were called across all samples (Methods). With a median of 92,782,803 reads per sample (range 21,545,569 to 182,140,303), sequencing depths were moderately correlated ($\rho = 0.487$) with the total genetic variants called per sample (Supplementary Fig. 1a).

To determine the efficiency of estimating genetic PCs from RNAseq data, we investigated the effects of minor allele frequency (MAF), linkage disequilibrium (LD) and the use of a pre-specified set of genetic variants. We found that the selection of a MAF threshold of >0.05 and a pairwise LD threshold of $r^2 < 0.05$ struck the optimal balance of offering the most variants for analysis and the highest correlation between RNAseq- and array-based genetic PCs (Supplementary Fig. 2). From the total of 4,921,472 genetic variants, 152,072 SNPs passed the MAF filter (MAF > 0.05), and 36,440 SNPs further passed the LD filter (LD < 0.05). Genetic variants from paired genomic data are available for 299 out of the initial 376 individuals; a total of 552,758 SNPs were identified and passed initial quality control filters (Methods), of which 315,615 SNPs and 29,943 SNPs then passed MAF > 0.05 and further LD < 0.05 filters, respectively. Out of the 299 samples with both RNAseq and paired array genotypes, 280 of them passed quality control and were used for further downstream analyses.

Among the 280 samples with matched array and RNAseq-based genotypes, we found 7343 overlapping SNPs between the MAF-filtered RNAseq and array SNP sets based on their exact chromosome positions. Genetic concordance was then calculated from the common SNPs based on matching allele genotypes for each position, taking into consideration strand flipping. Most RNAseq samples were found concordant with their respective paired array genotypes, with mean concordance of 0.925 for all samples and 232 sample-pairs (82.8%) having concordance higher than 0.90 (Fig. 2a). We found that high RNAseq depth was positively correlated with high genetic concordance with paired array genotypes; however, outliers of low genetic concordance were also present at high depths ($\rho = 0.1926$; Supplementary Fig. 1b).

When looking at the correlations between array-based genetic PCs and RG-PCs, we found that merely filtering based on MAF and LD was not adequate, as the main RG-PCs (specifically PCs 1 and 2) did not represent the genetic structure found in the array-based PCs, and meaningful correlation was only found from RG-PC3 and on (Supplementary Fig. 3). We found that the common approach of subsetting the genotype calls to the variants in HapMap3[25] offered higher quality genotype calls and improved correlation between RNAseq and array-based genetic PCs (Supplementary Fig. 4). An overlap of 23,227 well-defined SNPs was found between HapMap3 and the MAF-filtered (MAF > 0.05) variants, of which 4887 passed the LD filter (LD < 0.05) and were used to calculate RG-PCs. We also calculated genetic PCs from the 29,943 paired genotype array SNPs as a measure of true

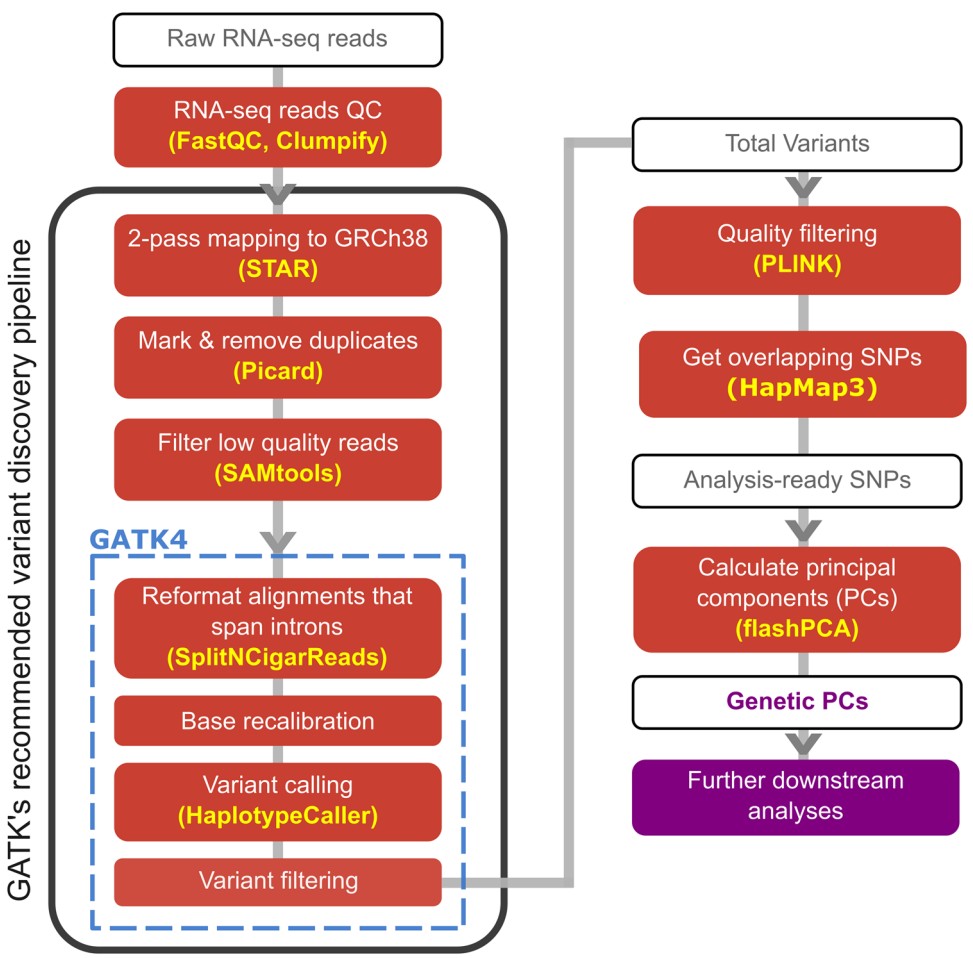

**Fig. 1 The analysis pipeline used in this study, mainly following GATK's recommended variant discovery pipeline to call genetic variants from RNAseq samples.** Further filtering based on missingness, MAF, and LD was then done before generating principal components (PCs) representing the population structure.

genetic structure to be compared against RG-PCs. To assess the consistency of inferred population structure between the two approaches, we calculated Spearman correlation between genetic PCs from paired genotype array SNPs and the RG-PCs. PC1 of both RNAseq and array sets correlated strongly with each other ($|\rho| = 0.93$), followed by RG-PC3 and PC2 from array data ($|\rho| = 0.61$) and RG-PC2 and PC3 from array data ($|\rho| = 0.6$) (Supplementary Fig. 4). As expected, the genetic PCs of one approach do not exclusively correspond to only one PC of the other approach, as can be seen with significant correlations of a single array PC with several RG-PCs. To investigate this further, we performed canonical correlation analysis between the top 10 array PCs and the RG-PCs and found that the RG-PCs fully explained the variance of the top 10 array PCs (Fig. 2b).

PCA of both array and RG-PCs showed visible clustering by self-reported patient ethnicity. Array PC1 vs PC2 captured the clustering of Janajati-Hill, Newar, and Madhesi groups, with array PC3 showing a clear distinction between the Newar samples and other samples (Fig. 3). Consistent with their Spearman correlations, RG-PC1, RG-PC2, and RG-PC3 also captured the clustering of the groups shown by the array data yet lacked some distance between the groups primarily due to array PC2.

**RNAseq-based variant calling can differentiate the genetic structure of distinct groups between and within populations akin to paired array genotypes.** We also tested RGStraP's

performance on a dataset of 465 samples from the Geuvadis consortium, spread across 5 different populations: British in England and Scotland (GBR), Utah residents with Northern and Western European ancestry (CEU), Finnish in Finland (FIN), Toscani in Italy (TSI), and Yoruba in Ibadan, Nigeria (YRI)[23]. A total of 463 samples passed the downstream filtering as part of the variant calling process (Methods). Clustering by each population can be seen in the main PCs from both paired array and RNAseq data: PC1 separates the European (EUR) and African (AFR) samples, while PC2 separates the EUR samples, with more distinct clusters seen for FIN and TSI samples (Supplementary Fig. 5A). Canonical correlation analysis between the top 10 array PCs and RG-PCs showed that RG-PCs fully explained the variance of the array PCs, with CV1 from RG-PCs representing 0.903 proportion of variance shared and the first 3 CVs ($R_{c1} = 0.994$, $R_{c2} = 0.942$, $R_{c3} = 0.752$) reaching a cumulative proportion of shared variance of 0.998 (Fig. 4a).

To project the Nepalese samples into PC space with other populations, we performed PCA of the Nepalese and Geuvadis samples together. Akin to the array PCs, results from RGStraP were able to distinguish the broad ancestry groups; RG-PC1 was able to separate between AFR and other samples, whereas RG-PC2 distinguished the EUR and Nepal (in this case, representing South Asian) samples (Fig. 4b). Separation within the EUR samples was also visible in RG-PC4, showing distinct clusters of FIN and TSI samples (Fig. 4b). When projected with the Geuvadis samples, clusters of the Nepalese self-reported ethnicity

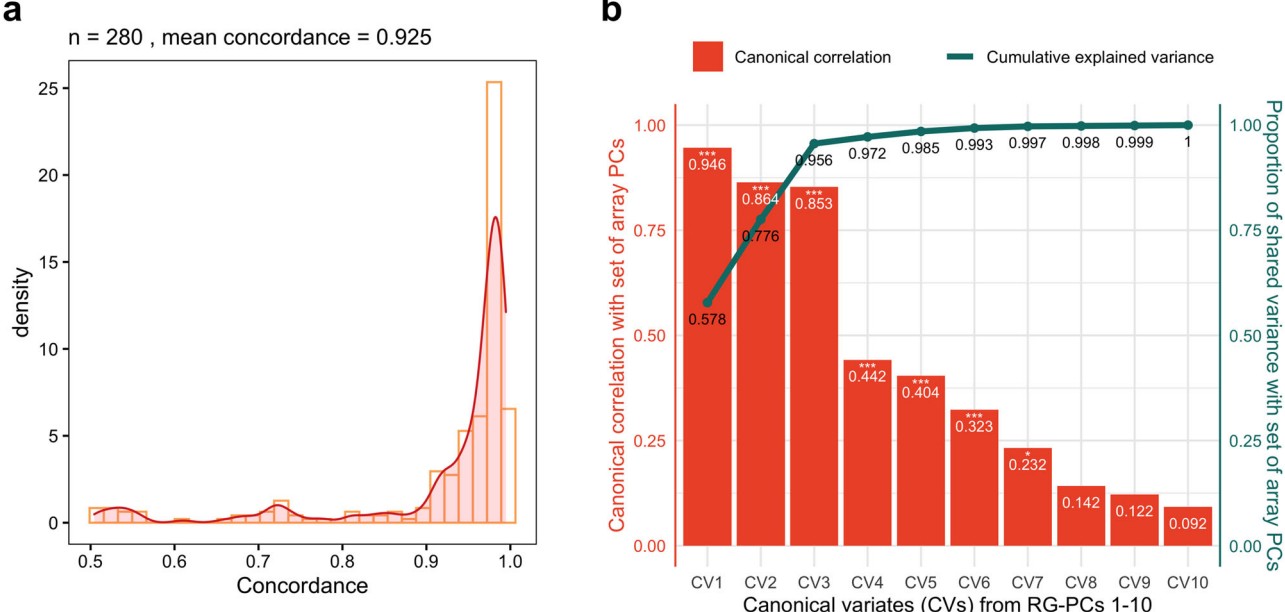

**Fig. 2 RGStraP was able to capture population structure in the Nepal cohort comparable to paired array genotype. a** Genotype concordance of common SNPs between array and RNAseq samples was found to be high, with most samples (232 out of 280) reaching >0.90 concordances. **b** Canonical correlation analysis between ten RG-PCs and ten array PCs showed significant (Wilks' Lambda, $p$-value < 0.05) correlations for the first 7 canonical variates (CVs) between the two sets. The first 3 CVs from 10 RG-PCs strongly captured the genetic information from array PCs ($R_{c1} = 0.946$, $R_{c2} = 0.864$, $R_{c3} = 0.853$), in which the cumulative proportion of shared variance between the two sets reached up to 0.956 from just the 3 CVs.

groups were also still visible (Supplementary Fig. 5b). This analysis further validates the performance of RGStraP in capturing genetic structure comparable to array genotypes across diverse populations, which is also supported by the canonical correlation analysis results (Supplementary Fig. 5c).

**RNAseq-based genetic PCs in differential gene expression analysis.** To assess the extent to which genetic PCs control population stratification in differential gene expression analysis (DGE) of these individuals, we performed DGE on the 280 Nepal samples to identify sex-specific gene expression, with and without adjustment for genetic PCs using either array PCs or RG-PCs (Methods). Prior to using edgeR for DGE analysis, we filtered out lowly expressed genes based on counts per million (CPM > 0.05) to account for differences in sequencing depths. Only autosomal genes were included in the analyses.

A systematic reduction was evident when adjusting for population structure in DGE analysis. A total of 3038 ($p$-value < 0.05) and 325 (FDR < 0.05) genes were differentially expressed when adjusting only for age and disease groups, whereas the number declined after adjusting with genetic PCs; inclusion of array PCs reduced the numbers of differentially expressed genes to 2585 ($p$-value < 0.05) and 144 (FDR < 0.05), while the number declined to 2778 ($p$-value < 0.05) and 272 (FDR < 0.05) when including RG-PCs. The majority of the differentially expressed genes identified without considering genetic PCs were still found after including either array PCs or RG-PCs (2478 and 2381 at $p$-value < 0.05, respectively; 138 and 213 at FDR < 0.05, respectively). The majority of DE genes were shared between the results using array PCs and RG-PCs (2175 at $p$-value < 0.05 and 130 at FDR < 0.05). When taking into consideration log-fold change, 4 genes passed the filter (FDR < 0.05, |logFC| > 1) in the set without considering genetic PCs, and the number decreased to 3 when including either array or RG-PCs. This demonstrates how RG-PCs control for population stratification in downstream RNAseq analysis similar to the

genetic PCs calculated from paired array genotypes, reducing significant associations that reflected variations in population structure instead of the biology of interest.

Effects of population structure were visible in quantile-quantile (QQ) plots, with attenuated test statistics across the transcriptome in the analysis with RG-PCs (Fig. 5a). This is quantitatively supported by lower systematic inflation ($m$); comparing the ratio of medians of the chi-square statistics between DGE results without genetic PCs and with RG-PCs as covariates, we found a slight systematic reduction of test statistics after including RG-PCs ($m = 0.935$). A similar reduction is also found when using array PCs as covariates ($m = 0.92$; Supplementary Fig. 6). Finally, we assessed a mixed linear model (MLM) with a genomic relationship matrix (GRM) constructed from the RNAseq-based SNPs (Methods). Similarly, we found that RG-PCs were able to control for population structure in the RNAseq-based GRM ($m = 0.985$) (Fig. 5b).

## Discussion

Population structure is typically captured via genotype arrays, which are not always done for projects focused on gene expression analyses. Genotyping may not be practical when resources are limited, or when analyzing existing and/or publicly available RNAseq datasets which nearly always do not offer access to original samples. In this study, we demonstrated how SNPs acquired solely from RNAseq variant calling were able to capture population structure comparable to the results from array data. We further showed that RNAseq-based genetic principal components (RG-PCs) were able to control for population structure in differential gene expression analysis, and that mixed linear model analysis using a genetic relatedness matrix based on RNAseq genotype calls was able to achieve similar. To facilitate the use of RG-PCs, we also construct and make freely available the tool RGStraP (https://github.com/fachrulm/RGStraP) for the wider research community.

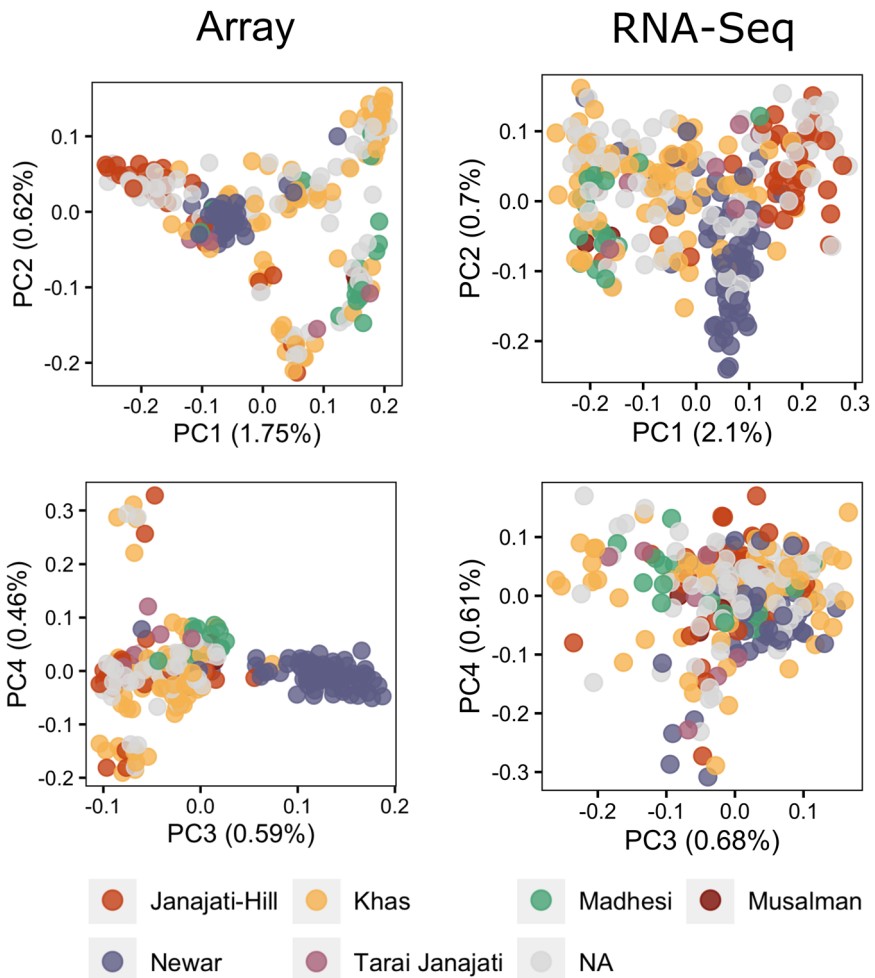

**Fig. 3 Population structure from both RGStrap and paired array genotype were consistent with self-reported ethnicities.** SNPs from the overlap of RNAseq variant calling results and HapMap3 were able to capture genetic structure separating self-reported ethnicities in Nepal when compared to the paired array genotypes, shown by the groupings in the PCA plots.

Our study enables the control of genetic population structure in analyses of current and historic transcriptomic datasets, which frequently do not come with matched genotypes. This is particularly important in low resource settings, in particular LMICs (e.g. Nepal) with highly diverse and structured populations or for traits where fine-scale genetic population structure may be confounding.

Our study has several limitations. The main challenge of constructing genetic PCs from RNAseq variant calling results is how to properly curate the SNPs to include, as MAF and LD filtering are not enough to remove uninformative SNPs, resulting in the main PCs not corresponding to self-reported ethnicity. This was addressed by using the overlapping SNPs between the RNAseq variant calling results and HapMap3 variants, allowing us to acquire a set of well-established SNPs; however, this SNP set may still miss important genetic structure, e.g. low-frequency and rare variants. Due to the nature of RNAseq platforms only capturing variants in transcribed regions, rarer variants that fall in crucial functional regions such as promoter and enhancer regions go unaccounted for[25]. These rare, non-transcribed variants are mostly linked to increased risk of various diseases and are enriched within expressed quantitative trait loci (eQTLs)[26–28].

Variants called from RNAseq data using a comparable GATK method have been reported to be reliable for tissue-specific eQTL mapping and allele-specific expression (ASE) analyses[19]. However, the approach heavily relied on genotype imputation using a reference panel that remains predominantly European, rendering the approach still suboptimal for eQTL analysis in other populations, especially those that are understudied. Thus, the method described in our study should not be utilized as a one-to-one replacement for conventional genotyping array or whole genome sequencing.

In summary, we have developed an approach and tool for inferring genetic population structure directly from RNAseq data across diverse populations, then demonstrated its use in differential gene expression analysis to control for genetic structure where genotyping data are unavailable. We hope that our results enable better control for confounding in RNAseq analyses and facilitate more rigorous retrospective and meta-analyses of RNAseq data.

## Methods

**Human ethics**. The STRATAA study was approved by the Nepal Health Research Council (NHRC, ref 306/2015) and OxTREC (Oxford Tropical Research Ethics Committee, ref 39-15). All participants provided informed consent for human genetic tests. Blood and nucleic acid samples, and associated data, were de-identified by the STRATAA team in Nepal prior to being sent to overseas for analysis.

**Sample processing, library preparation, and sequencing conditions**. This study utilized blood samples collected in Lalitpur, Nepal as part of the Strategic Typhoid Alliance across Africa and Asia (STRATAA) study, which included passive surveillance for enteric fever and a population-based serosurvey[29,30]. Blood was

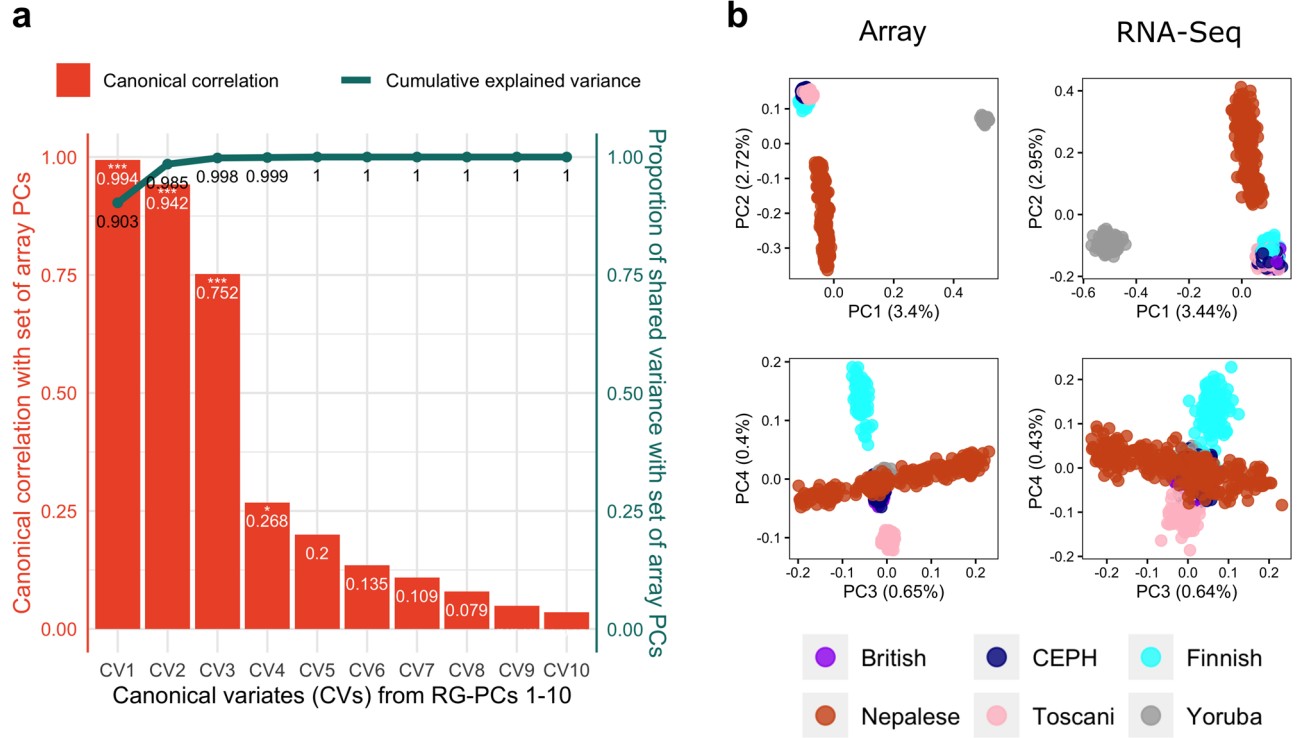

**Fig. 4 RGStraP was also tested on samples from the Geuvadis cohort, which includes 5 populations of European and African ancestries. a** Canonical correlation analysis between ten RG-PCs and ten array PCs of the Geuvadis samples showed significant (Wilks' Lambda, p-value < 0.05) correlations for the first 4 canonical variates (CVs) between the two sets. The first 3 CVs from 10 RG-PCs strongly captured the genetic information from array PCs ($R_{c1} = 0.994$, $R_{c2} = 0.942$, $R_{c3} = 0.752$). The cumulative proportion of shared variance between the three sets reached 0.998 from just the 3 CVs, 0.903 of them represented by CV1. **b** PCA plots of the Nepal and Geuvadis samples showing comparable population structure between the array-based PCs and RG-PCs, separating the ancestry groups (European, African, and South Asian) in the main PCs.

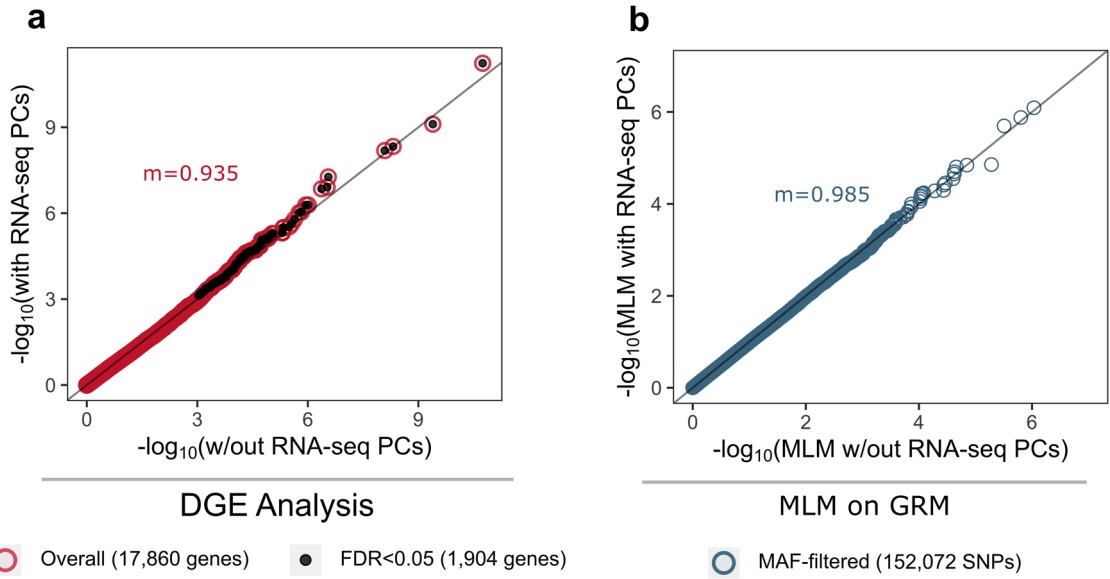

**Fig. 5 Quantile-quantile (Q-Q) plots showing the distribution of the probabilities between analyses with and without including genetic PCs.** Differential gene expression analysis results between samples of different sex show (**a**) systematic reduction in test statistics when including RNAseq-based genetic PCs as covariates compared to without, demonstrated by the low systematic inflation metric (m); **b** Distribution of probabilities after mixed linear model (MLM) on a genetic relationship matrix (GRM) was also found to be slightly deflated in the analysis with genetic PCs compared to the one without, though not to the extent of the DGE analysis.

collected from febrile participants recruited into the passive surveillance study, specifically those presenting >38.5 °C temperature or history of fever for >72 h. From the total blood sample volumes (≤16 mL for patients >16 years of age, ≤7 mL for ≤16 years), aliquots were subjected to (i) bacteriological culture to identify presence of *Salmonella enterica* serovars Typhi (*S.* Typhi); (ii) storage in PAXgene tubes for later RNA extraction; and (iii) DNA extraction and subsequent human genotyping. Blood was also collected from healthy participants in the serosurvey (≤8 mL for patients >16 years of age, ≤7 mL for ≤16 years), from which aliquots were also subjected to (i) serological analysis; (ii) PAXgene storage for RNA analysis; and (iii) DNA extraction.

We analysed RNA from 49 *S.* Typhi culture-positive participants and 275 *S.* Typhi culture-negative participants from passive surveillance, and 52 healthy controls from the serosurvey. PAXgene tubes were sent to Monash University in Melbourne, Australia, where RNA was extracted using PAXgene Blood RNA Kit. An aliquot of RNA (60 μl, 10–20 ng/ul) for each sample was sent to the Wellcome Sanger Institute in Hinxton, England for sequencing. Library preparation was done using NEBNext Ultra II RNA custom kit performed on an Agilent Bravo WS automation platform, pulling down poly(A) tails. After polymerase chain reaction (PCR) cycles (14 standard cycles), plates were purified using Agencourt AMPure XP SPRI beads and libraries were then quantified using Biotium Accuclear Ultra high sensitivity dsDNA Quantitative kit. Pooled libraries were quantified using Agilent bioanalyser and normalised to 2.8 nM. Samples were then subjected to globin depletion using the KAPA RNA HyperPrep with RiboErase (HMR) Globin kit. Libraries were subjected to 2 × 100 bp paired-end sequencing using the Illumina Novaseq platform. Depending on the level of multiplexing, each library was sequenced up to three times to generate an average of up to 80 million reads per sample.

For 299 of the Nepalese participants with RNAseq data, human DNA was successfully extracted (at Patan Hospital in Nepal) using the QIAamp DNA blood midi kit (QIAGEN) and DNA aliquot was shipped to the Genome Institute of Singapore for SNP genotyping. Genotyping was performed using the Illumina Infinium Global Screening Array-24 Kit (GSAMD-24v2_A1 chip). Stringent quality control filters were used to remove poorly performing SNP markers. SNPs were removed based on the following parameters; SNPs with a call rate of <95%; SNPs with differential call-rate between cases vs control $P$ value < 0.001; SNPs with a Hardy-Weinberg equilibrium $P$ value < 1e−07 in controls; SNPs with a Hardy-Weinberg equilibrium $P$ value < 1e−11 in cases. Samples with a call rate of <95%, those with a lower call rate sample from each sample pair having first- or second-degree relationship from identity by descent (IBD), and those that were PCA outlier samples were removed.

We also used a validation dataset from the Geuvadis consortium, which contains 465 lymphoblastoid cell line (LCL) samples from five of the 1000 Genomes populations: British in England and Scotland (GBR), Utah residents with Northern and Western European ancestry (CEU), Finnish in Finland (FIN), Toscani in Italy (TSI), and Yoruba in Ibadan, Nigeria (YRI)[23,31]. After extraction of total RNA with TRIzol Reagent (Ambion) and assessment of RNA quality with Agilent Bioanalyzer RNA 6000 Nano Kit, library preparation was done using TruSeq RNA Sample Prep Kit v2 for 2 × 75 bp paired-end mRNA sequencing on the Illumina HiSeq2000 platform[23]. The available paired genotype data was obtained from the 1000 Genomes Project Phase 3 dataset[31].

**Quality control and sequencing data processing.** Computational analyses were performed on an HPC cluster with 6 nodes, each equipped with 2 × 16 cores (32 threads) CPUs and 512GB memory. It is advised to run RGStraP on a high-performance computing (HC) cluster as the alignment step alone requires a minimum RAM of 32GB. We recommend at least 100GB of memory, a multi-core processor with at least 8 cores, and ample storage depending on the number of samples processed. A workload management system such as SLURM is also recommended for better resource management.

Quality control (QC) was done to the sequencing data according to the *FastQC* (bioinformatics.babraham.ac.uk/projects/fastqc) readouts. Illumina adapters were then trimmed using *Trim Galore!* (bioinformatics.babraham.ac.uk/projects/trim_galore), and optical duplicates were removed using bbmap's Clumpify. We then followed GATK's best practices pipeline for RNAseq short variant discovery, which commenced with a 2-pass mapping process to human genome GRCh38 using STAR[32], from which the resulting sequencing alignment map (BAM) files from the different runs of each sample were merged using Picard[33]. Low quality reads (MAPQ < 20) were filtered out from the merged files using SAMtools[34], followed by further QC, variant calling and filtering steps with GATK4[12]. The analysis-ready variants were then filtered using PLINK[35,36], removing duplicated and palindromic variants and only keeping autosomal single nucleotide polymorphisms (SNPs). We excluded individuals with more than 20% missing genotypes and further filtered RNAseq-based SNPs, after removing variants in the HLA region, based on minor allele frequency (MAF; $maf = 0.01$, $hwe = 0$, $geno = 0.1$, $mind = 0.2$) as well as linkage disequilibrium (LD; $win = 1000$, $step = 50$, $r2 = 0.05$). RG-PCs were successfully captured for 351 and 463 individuals in the Nepal and Geuvadis cohorts, respectively. The same SNP filters were also applied to the array-based SNPs, and for the SNP-level correlation analyses we converted genome annotations of the array SNPs from GRCh37 to GRCh38 (http://genome.ucsc.edu/cgi-bin/hgLiftOver). Merging of the SNP sets from the Nepal and Geuvadis samples was done using PLINK[35,36].

**Calculating concordance and visualizing stratification.** To calculate SNP-level concordance, we took overlapping SNPs between MAF-filtered array and RNAseq results based on chromosome position and calculated paired sample concordance based on matching genotype (considering heterozygosity and possible strand flipping) for each SNP in both results. A total of 280 Nepalese samples with RG-PCs and matching array genotype was used for this analysis.

To keep only meaningful SNPs from the RNAseq variant calling results for generating genetic principal components (PCs), we took overlapping SNPs (identical chromosome positions and matching genotypes taking into consideration possible strand flipping) between MAF-filtered RNAseq SNPs and HapMap3 variants to get a set of well-defined SNPs, after which LD filtering was performed on the set. We then generated genetic PCs for the LD-filtered RNAseq and array SNP sets separately using flashPCA[37], from which PCA plots were created with ggplot2[38] and Spearman correlations between the two sets of PCs were calculated using the function from the *stats* package of R[39]. We also calculated canonical correlation coefficients[40–42] between the array PCs and RG-PCs to assess how well the latter captured genetic structure information presented by the former, as a single array PC may be represented by multiple RG-PCs.

**Differential gene expression analysis between phenotypes.** Sequencing count data of the Nepalese samples were extracted from the aligned sequence files using *featureCounts* (http://bioinf.wehi.edu.au/featureCounts/), from which we kept only autosomal genes and genes with CPM > 0.05 in at least 20% of the samples from the analyses. Differential gene expression (DGE) analyses was done contrasting males and females using edgeR[43,44], taking into account age, disease group, and sequencing batches; we ran the analyses with and without populations structure PCs as an additional covariate to then compare how genetic structure may stratify gene expression. From both results, we also plotted the Q-Q plot and calculated the systematic inflation (m), which is the ratio of the median of the empirically observed chi-squared test statistics (in our case, results of DGE analysis with RG-PCs) to the expected median chi-squared test statistics (results of DGE analysis without RG-PCs), to quantify the stratification due to population structure in gene expression data.

We also created a genetic relationship matrix (GRM) and ran a mixed linear model (MLM) on the RNAseq SNPs with and without population structure PCs as random effects with fastGWA[45]. From the two results, a quantile-quantile (Q-Q) plot was created and systematic inflation (m) was then calculated to quantify the effect of populations structure PCs on the genomic data[46].

**Reporting summary.** Further information on research design is available in the Nature Portfolio Reporting Summary linked to this article.

## Data availability

Sequence data associated with this study were deposited to the European Genome-phenome Archive (EGA) under accession number EGAD00001011131. All other data supporting the findings are either presented as Supplementary Data 1 or may be obtained from the STRATAA study group (contact Dr Mila Shakya at milashakya@oucru.org).

## Code availability

The RGStraP pipeline is available at https://github.com/fachrulm/RGStraP and Zenodo at https://doi.org/10.5281/zenodo.8080230[24].

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

## Acknowledgements

We acknowledge the contributions of individuals and organizations who have arranged and taken part in the studies as well as the laboratory and field teams at the site, including the STRATAA Study Group and the Nepal Family Development Foundation team. We thank the Sanger sequencing teams. This research was funded in whole, or in part, by the Wellcome Trust [STRATAA, 106158/Z/14/Z and Sanger, 098051]. For the purpose of open access, the author has applied a CC BY public copyright licence to any Author Accepted Manuscript version arising from this submission. This research was also funded by NHMRC [project grant APP1101728] and supported by core funding from the British Heart Foundation (RG/18/13/33946) and the NIHR Cambridge Biomedical Research Centre (BRC-1215-20014; NIHR203312)[*]. *The views expressed are those of the author(s) and not necessarily those of the NIHR or the Department of Health and Social Care. M.I. is supported by the Munz Chair of Cardiovascular Prediction and Prevention and the NIHR Cambridge Biomedical Research Centre (BRC-1215-20014; NIHR203312) [*]. M.I. was also supported by the UK Economic and Social Research 878 Council (ES/T013192/1). M.F. was supported by a Melbourne Research Scholarship from The University of Melbourne jointly funded by the Baker Heart and Diabetes Institute. This work was supported by Health Data Research UK, which is funded by the UK Medical Research Council, Engineering and Physical Sciences Research Council, Economic and Social Research Council, Department of Health and Social Care (England), Chief Scientist Office of the Scottish Government Health and Social Care Directorates, Health and Social Care Research and Development Division (Welsh Government), Public Health Agency (Northern Ireland), British Heart Foundation and Wellcome. This study was also supported by the Victorian Government's Operational Infrastructure Support (OIS) program. *The views expressed are those of the authors and not necessarily those of the NIHR or the Department of Health and Social Care. The funders had no role in study design, data collection and analysis, decision to publish, or preparation of the manuscript. The views expressed in this manuscript are those of the author(s) and not necessarily those of the NIHR or the Department of Health and Social Care.

## Author contributions

M.F. and M.I. conceived the idea for RGStraP and wrote this manuscript. S.T., S.B., A.J.P., C.D., B.B., S.J.D., and K.E.H. are part of the core team of S.S.G. who contributed to the conception and design the main framework of several *S.* Typhi studies, including the main dataset used in this project. A.K., M.S., S.D., R.S., and B.B. are part of the Nepal team and were involved in the fieldwork design and execution in Nepal, including community engagement, logistics planning, and sampling, among other things. L.M.J., T.H., and K.E.H. were responsible for the Nepal RNAseq sample preparation, including RNA extraction and sequencing library preparation. C.C.K. and K.S.S. processed and pre-analysed the paired genotyping data of the Nepal samples. M.F. performed the computational experiments, analysed the data, prepared the figures, and constructed the pipeline highlighted in the study. M.I., S.J.D., and A.S. supervised the computational experiments. All authors read and approved the manuscript.

## Competing interests

C.C.K. is an Editorial Board Member for *Communications Biology*, but was not involved in the editorial review of, nor the decision to publish this article. All other authors declare no competing interests.

## Additional information

## STRATAA study group

Anup Adhikari[4], Happy Chimphako Banda[22], Christoph Blohmke[8], Thomas C. Darton[8], Yama Farooq[8], Maheshwar Ghimire[4], Jennifer Hill[8], Nhu Tran Hoang[23], Tikhala Makhaza Jere[22], Moses Kamzati[22], Yu-Han Kao[24], Clemens Masesa[22], Maurice Mbewe[22], Harrison Msuku[22], Patrick Munthali[22], Tran Vu Thieu Nga[23], Rose Nkhata[22], Neil J. Saad[24], Trinh Van Tan[23], Deus Thindwa[22], Farhana Khanam[25], James Meiring[8,22,26], John D. Clemens[25,27], Gordon Dougan[28], Virginia E. Pitzer[24], Firdausi Qadri[25], Robert S. Heyderman[29], Melita A. Gordon[22,30,31,32] & Merryn Voysey[8]

[22]Malawi-Liverpool Wellcome Programme, Blantyre, Malawi. [23]The Hospital for Tropical Diseases, Wellcome Trust Major Overseas Programme, Oxford University Clinical Research Unit, Ho Chi Minh City, Vietnam. [24]Department of Epidemiology of Microbial Diseases and the Public Health Modeling Unit, Yale School of Public Health, Yale University, New Haven, CT, USA. [25]International Centre for Diarrhoeal Disease Research, Dhaka, Bangladesh. [26]Department of Infection, Immunity and Cardiovascular Disease, University of Sheffield, Sheffield, UK. [27]International Vaccine Institute, Seoul, South Korea. [28]Department of Medicine, Cambridge Institute of Therapeutic Immunology and Infectious Diseases (CITIID), University of Cambridge, Cambridge, UK. [29]National Institute for Health Research Global Health Research Unit on Mucosal Pathogens, Division of Infection and Immunity, University College London, London, UK. [30]Institute of Infection, Veterinary & Ecological Sciences, University of Liverpool, Liverpool, UK. [31]Kamuzu University of Health Sciences, Blantyre, Malawi. [32]Department of Clinical Sciences, Liverpool School of Tropical Medicine, Liverpool, UK.

