## [Peer Review File · Communications Biology]

Reviewers' comments:

Reviewer #1 (Remarks to the Author):

Fachrul et al present a pipeline for estimation of genetic principal components (PCs) from genotypes inferred from RNA-seq data. Overall, the study is well-motivated, clearly described and the conclusions are supported by the data presented. This tool fills an unmet need as genetic population structure has been clearly demonstrated in RNA sequencing data but genotype data are not always available, and practitioners do not always consider correcting for this structure. However, I do have a few minor comments I would like to see addressed:

- The authors claim that subsetting the SNPs to HapMap3 improves correlation with the array-inferred PCs (Figure 3, Figure 2B). However, unless I am missing it, I don't see any numbers presented for these analyses without subsetting to HM3. What do Figure 3 and Figure 2B look like without subsetting to HM3?
- Along those lines, subsetting to HM3 is a reasonable strategy for obtaining a set of ancestry-informative SNPs, but given that this is an understudied population I am wondering what the relationship between this Nepalese population and the HM3 populations is. For example: where would the Nepalese genotypes project into HM3 PC space? - It may be worth addressing this point in the discussion as we need broader, more diverse global reference panels to continue to improve biomedical research in these populations.
- In the discussion around Figure 4, how do these inflation statistics corrected using RNA PCs compare to using array PCs? I would expect it to be slightly worse, which is acceptable, I'd just like to see that comparison quantified.
- L171 mentions correcting for disease groups, but this is the first mention of any disease groups within this study population. This is discussed more in the methods, but you should mention the existence of disease status in this study earlier, perhaps in the introduction.
- In figure S2, it looks like the < and > are flipped in the margin.

Reviewer #2 (Remarks to the Author):

In this manuscript, the authors build upon previous work that calls genotypes from RNA-sequencing data, and estimate principal components from these genotypes. They have developed a pipeline allowing users to estimate RNA-seq principal components as a proxy for population structure, allowing for correction without having to additionally perform genotyping using genotyping arrays. They demonstrate the performance of their method on a novel RNA-sequencing dataset of blood samples from various groups in Nepal, provide optimal parameters for PC estimation, and quantify improvement in accuracy of differential gene expression results when correcting for these principal components.

While the use of RNA-Seq-derived PCs to estimate population structure has been demonstrated previously (Deelen et al, 2015: please include this paper in your citations), there is no existing tool to implement this automatically. An easily-implementable package would be a handy tool that can be applied to a host of differential expression studies, and also retrospectively to published studies to estimate population structure effects. The manuscript is very clearly written, and was a pleasure to read.

However, given that the primary focus of the paper is on a new tool and not the dataset being described, more applications of the authors' methods on existing population-based datasets would make the paper more complete (listed in the comments). Moreover, adding more flexibility to the

github package would make its functionality more generalizable. These suggestions are expanded on below:

Some comments for the authors:

1. Github software package:

1a. Since the primary message of this paper is their method and the RGStrap tool, it needs to be documented in more detail. A folder with an example config.yaml file, and example input data (including an example metadata file) would help set up the tool.

1b. It would be very helpful to have the option for users to start the pipeline after variants have been called and filtered. This would be especially helpful in post-hoc analysis of previous data which have already undergone variant calling.

2. Could the authors demonstrate the consistency of the proposed variant filtering parameters (MAF > 0.05; $r^2 < 0.05$) on other populations with the 1000Genomes/GTEX samples, where we know what to expect from PC correction? Similarly, demonstrating the use of RG-PCs in a previously-published gene expression dataset would help evaluating its use.

3. In the differential expression analysis, the number of RG-PCs identified and used for correction in the DE analysis is not clear. How many RG-PCs were used in the CCA with the top array-derived PCs? The authors report the numbers of DE genes identified with each analysis--are the DE genes from the array-based PC analysis a superset of the DE genes from the RNA-Seq-derived PC analysis?

Minor comments:

1. How many samples had paired array and RNA-seq based genotypes? Line 123 says 299, and line 128 says 280, and figure S1A says 284.

2. How would the differential gene expression P-values change if you used self-reported ancestry instead?

Point-by-point response to reviewers

Reviewer 1

General comments:

Fachrul et al present a pipeline for estimation of genetic principal components (PCs) from genotypes inferred from RNA-seq data. Overall, the study is well-motivated, clearly described and the conclusions are supported by the data presented. This tool fills an unmet need as genetic population structure has been clearly demonstrated in RNA sequencing data but genotype data are not always available, and practitioners do not always consider correcting for this structure. However, I do have a few minor comments I would like to see addressed:

Minor comments:

- The authors claim that subsetting the SNPs to HapMap3 improves correlation with the array-inferred PCs (Figure 3, Figure 2B). However, unless I am missing it, I don't see any numbers presented for these analyses without subsetting to HM3. What do Figure 3 and Figure 2B look like without subsetting to HM3?

Our response: We have added supplementary figures (now Figure S3) to demonstrate the noise issue without overlapping with HapMap3 SNPs, as the main PCs (PCs 1-2) do not represent the structure found on the paired array genotype. The population structure could only be inferred starting from PC3 if we do not subset the SNPs. This was amended on **lines 145-148 and Figure S3**.

- Along those lines, subsetting to HM3 is a reasonable strategy for obtaining a set of ancestry-informative SNPs, but given that this is an understudied population I am wondering what the relationship between this Nepalese population and the HM3 populations is. For example: where would the Nepalese genotypes project into HM3 PC space? - It may be worth addressing this point in the discussion as we need broader, more diverse global reference panels to continue to improve biomedical research in these populations.

Our response: We have tested RGStrAP using Geuvadis samples, which are spread across HM3 / 1000 Genomes samples from 5 different populations (British, Europeans in Utah, Finnish, Toscani, and Yoruba). We projected the Nepal samples into the Geuvadis PC space and found that Nepal samples were distinguishable from the populations, as the main PCs clearly separate the African, European, and South Asian (Nepalese) samples. This consistent separation is found in both array-based PCs and RG-PCs. We have amended the text related to this matter on **lines 174-197** and the **new Figures 4 and S5**.

- In the discussion around Figure 4, how do these inflation statistics corrected using RNA PCs compare to using array PCs? I would expect it to be slightly worse, which is acceptable, I'd just like to see that comparison quantified.

Our response: We have added a new figure for this (**Figure S6**), showing that using array PCs returns a similar degree of reduction in test statistics. We refer to this on **lines 229-230**.

- L171 mentions correcting for disease groups, but this is the first mention of any disease groups within this study population. This is discussed more in the methods, but you should mention the existence of disease status in this study earlier, perhaps in the introduction.

Our response: We thank the Reviewer 1 for pointing this out and have amended the text to mention this on **lines 111-113**.

- In figure S2, it looks like the < and > are flipped in the margin.

Our response: We thank Reviewer 1 for pointing this out and have amended the figure.

Reviewer 2

General comments:

In this manuscript, the authors build upon previous work that calls genotypes from RNA-sequencing data, and estimate principal components from these genotypes. They have developed a pipeline allowing users to estimate RNA-seq principal components as a proxy for population structure, allowing for correction without having to additionally perform genotyping using genotyping arrays. They demonstrate the performance of their method on a novel RNA-sequencing dataset of blood samples from various groups in Nepal, provide optimal parameters for PC estimation, and quantify improvement in accuracy of differential gene expression results when correcting for these principal components.

While the use of RNA-Seq-derived PCs to estimate population structure has been demonstrated previously (Deelen et al, 2015: please include this paper in your citations), there is no existing tool to implement this automatically. An easily-implementable package would be a handy tool that can be applied to a host of differential expression studies, and also retrospectively to published studies to estimate population structure effects. The manuscript is very clearly written, and was a pleasure to read.

Our response: We thank Reviewer 2 for the suggestion and have incorporated references to this paper on **lines 81-82 and 266-272**.

However, given that the primary focus of the paper is on a new tool and not the dataset being described, more applications of the authors' methods on existing population-based datasets would make the paper more complete (listed in the comments). Moreover, adding more flexibility to the github package would make its functionality more generalizable. These suggestions are expanded on below:

1. Github software package:

1a. Since the primary message of this paper is their method and the RGStrap tool, it needs to be documented in more detail. A folder with an example config.yaml file, and example input data (including an example metadata file) would help set up the tool.

Our response: We thank Reviewer 2 for the suggestion and have included updated config.yaml files for the main configurations, as well as an updated example for the slurm profile for the pipeline.

1b. It would be very helpful to have the option for users to start the pipeline after variants have been called and filtered. This would be especially helpful in post-hoc analysis of previous data which have already undergone variant calling.

Our response: We thank the reviewer for the suggestion and have provided a 'lite' version on the GitHub page for this purpose.

2. Could the authors demonstrate the consistency of the proposed variant filtering parameters (MAF > 0.05; $r^2 < 0.05$) on other populations with the 1000Genomes/GTEx samples, where we know what to expect from PC correction? Similarly, demonstrating the use of RG-PCs in a previously-published gene expression dataset would help evaluating its use.

Our response: As suggested, we have applied the pipeline on the Geuvadis samples and validated its utility, as the RG-PCs resulted in comparable population structures when compared to the paired array genotype data for those samples. The inclusion of the Geuvadis samples means we have tested the method and variant filtering parameters across different broad ethnic groups (African, European, and South Asian). This new addition is amended on **lines 174-197**, accompanied by the **new Figures 4 and S5**.

3. In the differential expression analysis, the number of RG-PCs identified and used for correction in the DE analysis is not clear. How many RG-PCs were used in the CCA with the top array-derived PCs? The authors report the numbers of DE genes identified with each analysis--are the DE genes from the array-based PC analysis a superset of the DE genes from the RNA-Seq-derived PC analysis?

Our response: CCA was done using 10 RG-PCs and 10 array PCs. The DE genes from including array PCs are not a superset of the results with RG-PCs, though the majority of DE genes were shared between the two sets. We have amended this on **lines 215-216**.

Minor comments:

1. How many samples had paired array and RNA-seq based genotypes? Line 123 says 299, and line 128 says 280, and figure S1A says 284.

Our response: Out of the initial 376 samples, 299 of them had paired array genotypes. The 280 samples we used for the concordance analyses were those that passed QC (overlap between 362 QCed RNAseq samples and 299 paired array genotype samples, from which outliers were removed based on PCA). We have amended the text on **lines 116-117, 128-133**, and **Figure S1** for clarity.

2. How would the differential gene expression P-values change if you used self-reported ancestry instead?

Our response: One of the main initiating factors driving this study is the fact that out of 375 samples, 108 of them (28.8%) lacked any self-reported ethnicity and 76 of them (20%) lacked paired array genotypes. Out of the 267 with self-reported ethnicity, only 206 of them had paired array genotypes, reducing the number of confirmed *S. Typhi* cases in less than half, from 52 to merely 22 samples. With how the PCA plots show clustering based on self-reported ethnicity, using self-reported ethnicity will reduce significant associations to a similar extent with array PCs and RG-PCs. We decided not to pursue the comparison as paired array genotypes serve as a superior point of comparison / ground truth for the RG-PCs, and that the loss of samples will reduce the power of the analysis.

REVIEWERS' COMMENTS:

Reviewer #1 (Remarks to the Author):

The authors have addressed all of my comments.

Reviewer #2 (Remarks to the Author):

The authors have addressed all the reviewers' comments satisfactorily. I have no further comments.

Point-by-point response to reviewers

Reviewer 1

General comments:

The authors have addressed all of my comments.

Our response: We thank Reviewer 1 for their helpful and constructive comments.

Reviewer 2

General comments:

The authors have addressed all the reviewers' comments satisfactorily. I have no further comments.

Our response: We thank Reviewer 2 for their helpful and constructive comments.